# Identifying Mental Health Literacy as a Key Predictor of COVID-19 Vaccination Acceptance among American Indian/Alaska Native/Native American People

**DOI:** 10.3390/vaccines11121793

**Published:** 2023-11-30

**Authors:** Xuewei Chen, Carrie Winterowd, Ming Li, Gary L. Kreps

**Affiliations:** 1School of Community Health Sciences, Counseling and Counseling Psychology, Oklahoma State University, Stillwater, OK 74078, USA; carrie.winterowd@okstate.edu; 2Department of Health Sciences, College of Health Professions, Towson University, Towson, MD 21252, USA; mli@towson.edu; 3Center for Health and Risk Communication, George Mason University, Fairfax, VA 22030, USA; gkreps@gmu.edu

**Keywords:** COVID-19, vaccine hesitancy, mental health, health literacy

## Abstract

Background: This study examines how health literacy and mental health literacy associate with the willingness to receive a COVID-19 vaccination among American Indian/Alaska Native/Native American (AI/AN) people. Methods: The data were collected with an online Qualtrics survey in February 2021 (*n* = 563). A purposive snowball sampling strategy was used by sending recruitment flyers to colleagues and organizations who work with AI/AN communities to share with appropriate potential respondents. We performed linear regression analyses examining the relationships between the willingness to receive a COVID-19 vaccination and socio-demographic characteristics such as age, gender, education, health literacy, mental health literacy, self-rated physical and mental health status, worry about getting COVID-19, perceived COVID-19 susceptibility, and perceived COVID-19 severity. Results: Mental health literacy and health literacy predicted 30.90% and 4.65% of the variance (R^2^_adjusted_) in the willingness to receive a COVID-19 vaccine, respectively. After holding the self-rated physical/mental health status, worry about getting COVID-19, perceived susceptibility, perceived severity, health literacy, and socio-demographics constant, mental health literacy was still a strong predictor (b = 0.03, *p* < 0.001) for the willingness to receive a COVID-19 vaccine (model R^2^_adjusted_ = 40.14%). Conclusions: We identified mental health literacy as a substantial factor associated with the willingness to receive a COVID-19 vaccination among AI/AN respondents.

## 1. Introduction

American Indian/Alaska Native/Native American (AI/AN) consists of individuals with origins among the indigenous peoples of North, South America, and Central America, who maintain tribal affiliation or community attachment [1]. The term “Native American” gained widespread usage in the 1970s as a substitute for “American Indian” in the United States (U.S.) [2]. In 2020, AI/AN comprised 9.7 million people, constituting 2.9% of the total U.S. population [1]. AI/AN people have experienced substantial health disparities during the COVID-19 pandemic. For example, AI/AN communities reported disproportionately higher rates of COVID-19, COVID-19-associated hospitalizations, and COVID-19-related mortality rates compared to other racial/ethnic majority groups [3,4,5]. Recently, researchers have found that the prevalence of COVID-19 vaccine hesitancy was highest among AI/AN respondents [6,7]. These findings can certainly be understood within the historical and cultural contexts of AI/AN people’s distrust of colonized ways and White medicine [8]. The primary purpose of the present study was to explore the factors associated with AI/AN people’s willingness to receive a COVID-19 vaccination, including their health literacy, their mental health literacy, self-reported health and mental health status, and some socio-demographic factors.

According to Healthy People 2030, personal health literacy is “the degree to which individuals have the ability to find, understand, and use information and services to inform health-related decisions and actions for themselves and others” [9,10]. The AI/AN population may have an increased risk for limited health literacy because of low educational attainment and high rates of poverty in many AI/AN communities [11]. According to the National Assessment of Adult Literacy, AI/AN people had lower average health literacy than White people [12,13]. Compared to people with higher health literacy, those with lower health literacy are more likely to trust and use vaccine misinformation for decision making [14,15], which may lead to lower vaccine confidence and negatively affects the vaccine uptake. Research has shown that people with lower health literacy levels were more likely to report COVID-19 vaccine hesitancy compared to those with higher health literacy [16,17] because those with limited health literacy had difficulty obtaining and understanding health information about the vaccine [18]. Thus, we hypothesize that low health literacy is a significant barrier to COVID-19 vaccine acceptance among AI/AN people.

In a prior study, people who were COVID-19 vaccine hesitant were less inclined to seek treatment for mental health concerns [19]. In another recent study [20], the association between health literacy and COVID-19 vaccine hesitancy was observed solely among individuals reporting low or moderate stress levels; among those who reported high levels of stress, the impact of health literacy on reducing COVID-19 vaccine hesitancy became inconspicuous. This finding is likely to relate to the powerful situational state conditions that patients often experience when seeking care, such as feeling ill, nauseous, dizzy, or having pain, that can seriously degrade their levels of health literacy [21]. The COVID-19 pandemic could intensify the manifestations of existing mental health issues and provoke the emergence of new ones, especially among AI/AN people because they are likely to experience a significantly elevated prevalence of mental health disorders [22]. These studies suggest that besides health literacy, mental health literacy might also play a significant role in predicting COVID-19 vaccine hesitancy among AI/AN people. Mental health literacy refers to “knowledge and beliefs about mental disorders with aid in their recognition, management or prevention” [23]. Poor mental health literacy is linked to challenges in recognizing mental health symptoms, identifying suitable treatment options, and is correlated with elevated levels of depression, anxiety, stress, and internalized stigma [24].

Low health literacy and low mental health literacy have been suggested as barriers to the COVID-19 vaccine uptake among the general population [25,26]. However, few researchers have investigated the contributions of health literacy and mental health literacy in predicting COVID-19 vaccination acceptance, specifically among AI/AN people. Therefore, the purpose of this study is to examine how health literacy and mental health literacy are associated with the willingness to receive a COVID-19 vaccination among American Indian/Alaska Native/Native American (AI/AN) people.

In addition to health literacy and mental health literacy, previous studies have linked COVID-19 vaccine hesitancy with various variables. Researchers found that COVID-19 vaccine hesitancy is associated with socio-demographic characteristics, being worried about getting COVID-19, as well as perceived COVID-19 susceptibility and severity [27,28,29,30]. For example, studies reported variation in COVID-19 hesitancy rates when examining groups by gender, age, and education attainment [31,32]. A better self-rated health status was associated with higher COVID-19 vaccine acceptance [33,34,35]. Worry about getting a disease, perceived susceptibility, and perceived severity are important constructs from the Health Belief Model [36]. Numerous studies have documented that these constructs are important predictors for the willingness to receive a COVID-19 vaccine. For example, those who were more worried about getting COVID-19 and had higher perceived severity of COVID-19 tended to have a stronger intention to receive a COVID-19 vaccine [33]. Higher perceived susceptibility and severity of COVID-19 were associated with lower COVID-19 vaccine hesitancy [37]. Therefore, we posed the following research questions:(1)Were AI/AN participants’ self-rated physical and mental health status, health beliefs related to COVID-19 (i.e., worry about getting COVID-19, their perceived COVID-19 susceptibility, and perceived severity of COVID-19 consequences), and socio-demographics (i.e., gender, age, and education) associated with their level of willingness to receive a COVID-19 vaccination?(2)Were AI/AN people’s health literacy and mental health literacy associated with their willingness to receive a COVID-19 vaccination?(3)Were AI/AN people’s health literacy and mental health literacy associated with their willingness to receive a COVID-19 vaccination, when accounting for self-rated physical and mental health status, health beliefs regarding COVID-19 (i.e., worry about getting COVID-19, perceived COVID-19 susceptibility, and perceived COVID-19 severity), and socio-demographics (i.e., gender, age, and education) as covariates?

## 2. Methods

### 2.1. Data Collection Procedure

We collected survey data for this study in February 2021. Our participation selection criteria included being (a) 18 years or older, (b) self-identified as American Indian/Alaska Native/Native American (AI/AN), and (c) physically located in the United States at the time when filling out the survey. We applied a purposive snowball sampling strategy by sending recruitment flyers to colleagues who conduct research and/or practice among AI/AN communities and professional/student organizations such as state Tribal Engagement Partners, the Native American Student Association (NASA) at the university, and the Institute for Healthcare Advancement Health Literacy Solutions Center. Those students and professionals further disseminated the recruitment information to AI/AN people within their social/professional networks, such as Facebook Groups. An anonymous link and a QR code to our online Qualtrics survey was included in the recruitment flyer. Each participant received a $10 Amazon electronic gift card as an incentive after completing the survey.

A total of 39 responses were removed from the datafile due to missing responses for all items, straight line responses, speed responses, and/or not passing either of the two validation check questions (i.e., asking participants to select “somewhat agree” for one item and “somewhat disagree” for the other item). Thus, our final sample size for the data analysis included 563 AI/AN participants. This study was approved by the Oklahoma State University Institutional Review Board.

### 2.2. Measures

#### 2.2.1. Outcome Variable: Willingness to Receive a COVID-19 Vaccination

We measured AI/AN participants’ willingness to receive a COVID-19 vaccination using a single item adopted from a prior study [38], asking participants how much they agree with the following statement, “I will take a COVID-19 vaccine if it is proven safe and effective and is available to me”. Participants used a five-point Likert scale to rate their response, from Strongly Disagree (coded as 1) to Strongly Agree (coded as 5). A higher score indicated a greater willingness to receive a COVID-19 vaccine.

#### 2.2.2. Predictor Variable: Health Literacy

We adapted an objective test, the Newest Vital Sign (NVS) [39] to measure participants’ general health literacy level. The NVS has been validated and demonstrates satisfactory reliability with various populations in the U.S. and across the globe [15,39,40]. The NVS is an objective test wherein individuals were tasked with interpreting a simulated ice cream nutrition label and responding to six open-ended questions [39]. Responses to each question were scored as either correct (coded as 1) or incorrect (codes as 0). Participants were provided with the option to leave the answer text box blank if they did not know the answer. Those leaving the answer text box blank or typed information such as “I don’t want to answer” and “na” were scored as 0. The Cronbach’s alpha coefficient (0.91) revealed strong internal consistency within the NVS. Higher sum scores indicated better health literacy levels. The possible NVS health literacy sum score range was 0 to 6.

#### 2.2.3. Predictor Variable: Mental Health Literacy

We adapted Jung and colleagues’ (2016) multi-component mental health literacy measure. This mental health literacy measure has been found to yield reliable and valid data among different populations in the U.S., including with college students and racial ethnic minority groups such as African Americans and Hispanics [41,42]. This measure [41] assesses individuals’ mental health literacy in three domains: knowledge-oriented (12 items), beliefs-oriented (10 items), and resource-oriented (four items). The knowledge-oriented items ask participants if they agree with statements, such as “Counseling/therapy is helpful for people with mental health needs” and “A person with schizophrenia may see things that are not really there” (yes, no, or don’t know). We coded “yes” as 1 and “no” or “don’t know” as 0.

The beliefs-oriented items (e.g., “A highly spiritual/religious person does not develop mental health concerns”) and resource-oriented items (e.g., “I know where to go to receive mental health services”) ask participants how much they agree with each statement, using a five-point Likert scale from Strongly Agree (coded as 1) to Strongly Disagree (coded as 5). Cronbach’s alpha coefficients exhibited good internal consistency of their overall mental health literacy scores and their subscales (0.91 for overall scores, 0.83 for knowledge-oriented scores, 0.90 for beliefs-oriented scores, and 0.73 for resource-oriented scores). We calculated a sum score by adding up all domains. Higher scores indicated better mental health literacy. The possible mental health literacy sum score range was 14 to 82.

#### 2.2.4. Covariates: Socio-Demographics, Self-Rated Health Status, Worry about Getting COVID-19, Perceived COVID-19 Susceptibility, and Perceived COVID-19 Severity

Given the previous research regarding these variables and their relationship to COVID-19 vaccine hesitancy ([27,28,29,30], we included the following variables as covariates in this study. The socio-demographic variables included gender (man, woman, transgender man, transgender woman, non-binary/third gender, other, and prefer to not disclose), age, and education (less than 8 years, 8 through 11 years, 12 years or completed high school, post-high school training other than college, some college, college graduate, and postgraduate).

We assessed participants’ self-rated physical and mental health status by asking, “In general, would you say your physical/mental health is…”, on a five-point Likert scale, from Poor (coded as 1) to Excellent (coded as 5).

We also asked participants to rate how much they were worried about getting COVID-19 on a five-point Likert scale, from Not At All (coded as 1) to Extremely (coded as 5), and their perceived chance of getting COVID-19 in their lifetime on a six-point Likert scale, from Extremely Unlikely (coded as 1) to Extremely Likely (coded as 6).

We examined participants’ perceived severity of getting infected with COVID-19, using five items which were rated on a five-point Likert scale from Strongly Disagree (coded as 1) to Strongly Agree (coded as 5). These items were modified from a prior study that focused on assessing illness perception [43]. Example items included “If I had COVID-19, it would have major consequences on my life”, “If I had COVID-19, it would have serious financial consequences”, and “If I had COVID-19, it would cause difficulties for those who are close to me”. The scores on our COVID-19 perceived severity scale exhibited acceptable internal consistency among our participants (Cronbach’s alpha = 0.71). We calculated the sum score for these five items and higher sum scores indicated higher perceived severity of getting COVID-19. The possible sum score range was 5 to 25.

### 2.3. Data Analysis

We performed sets of bivariate linear regressions to examine the relationships between the willingness to receive a COVID-19 vaccination and health literacy/mental health literacy, as well as between the willingness to receive a COVID-19 vaccination and each of following variables: socio-demographics (i.e., gender, age, and education), self-rated physical health status, self-rated mental health status, worry about getting COVID-19, perceived COVID-19 susceptibility, and perceived COVID-19 severity.

We also performed a multiple linear regression model to further clarify the roles of health literacy and mental health literacy in predicting the willingness to receive a COVID-19 vaccination among AI/AN people, while also statistically holding the self-rated physical/mental health status, worry about getting COVID-19, perceived susceptibility, perceived severity, and socio-demographics (i.e., gender, age, and education) constant. Multicollinearity was measured using a variance inflation factor (VIF). We used Stata 16 for statistical analysis. The significance level was set at α = 0.05.

## 3. Results

### 3.1. Sample Demographic Characteristics

Table 1 presents the demographic characteristics of the 563 AI/AN participants. In our sample, 347 of the respondents were men (62%), and 215 were women (38%). Their age ranged from 20 to 66 years old (M = 33.81, *SD* = 5.45). More than half of the AI/AN participants reported having some college education or a college degree (some college education, *n* = 247, 44%; graduated from college, *n* = 117, 21%).

### 3.2. Willingness to Receive a COVID-19 Vaccination

More than half of our participants (*n* = 310, 55%) indicated that they “agree” (*n* = 174, 31%) or “strongly agree” (*n* = 136, 24%) to the statement, “I will take a COVID-19 vaccine if it is proven safe and effective and is available to me”. About 17% of our participants (*n* = 98) chose “disagree” (*n* = 87, 15%) or “strongly disagree” (*n* = 11, 2%) to this statement. About 28% (*n* = 155) chose “neutral” to this statement. The mean score of one’s willingness to receive a COVID-19 vaccination was 3.60 (*SD* = 1.07). AI/AN women were less likely to be willing to receive a COVID-19 vaccination compared to men (b = −0.49, *p* < 0.001). Being older was associated with more willingness to receive a COVID-19 vaccine (b = 0.05, *p* < 0.001). Education level was not associated with the willingness to receive a COVID-19 vaccine (b = 0.06, *p* = 0.084).

### 3.3. Health Literacy

The NVS health literacy total score ranged from 0 to 6 (M = 1.96, *SD* = 2.26). More than half of the participants (54%) received a 0 score, indicating they did not know the answers to the NVS test or refused to answer the questions. About 39% of the participants received scores of 4–6, indicating they had adequate health literacy. Women reported lower NVS health literacy scores than men (b = −0.44, *p* < 0.001). Being older (b = 0.12, *p* < 0.001) and having a higher education (b = 0.08, *p* = 0.018) were associated with higher NVS health literacy scores.

### 3.4. Mental Health Literacy

The mental health literacy total score ranged from 32 to 82 (M = 53.10, *SD* = 12.79). AI/AN women reported lower mental health literacy than men (b = −8.13, *p* < 0.001). Being older (b = 0.80, *p* < 0.001) and having a higher education (b = 1.41, *p* = 0.001) were associated with better mental health literacy.

### 3.5. Self-Rated Physical Health Status

Most of our participants (*n* = 476, 85%) rated their physical health as “good” (*n* = 199, 35%), “very good” (*n* = 158, 28%), or “excellent” (*n* = 119, 21%). About 15% (*n* = 87) of the participants indicated that they had “fair” (*n* = 75, 13%) or “poor” (*n* = 12, 2%) physical health. The mean score of their self-rated health status was 3.52 (*SD* = 1.03). AI/AN women reported a poorer self-rated physical health status compared to men (b = −0.44, *p* < 0.001). Being older (b = 0.03, *p* < 0.001) and having a higher education (b = 0.08, *p* = 0.018) were associated with a better self-rated physical health status.

### 3.6. Self-Rated Mental Health Status

Most of our participants (*n* = 436, 77%) rated their mental health as “good” (*n* = 134, 24%), “very good” (*n* = 178, 31%), or “excellent” (*n* = 124, 22%). About 22.5% (*n* = 127) of the participants indicated that they had “fair” (*n* = 103, 18%) or “poor” (*n* = 24, 4%) mental health. The mean score of their self-rated mental health status was 3.49 (*SD* = 1.15). AI/AN women reported a poorer self-rated mental health status compared to men (b = −0.65, *p* < 0.001). Being older (b = 0.04, *p* < 0.001) and having higher education (b = 0.11, *p* = 0.005) were associated with a better self-rated mental health status.

### 3.7. Worry about Getting COVID-19

The majority of our participants (*n* = 466, 82%) indicated that they were “somewhat” (*n* = 119, 21%), “moderately” (*n* = 199, 35%), or “extremely” (*n* = 148, 26%) worried about getting COVID-19. About 2% (*n* = 11) of the participants indicated they were “not at all” and 15% (*n* = 85) indicated “slightly” worried about getting COVID-19. The mean score of worry about getting COVID-19 was 3.69 (*SD* = 1.08). Older age was associated with being more worried about getting COVID-19 (b = 0.03, *p* = 0.001). Gender (b = −0.16, *p* = 0.097) and education (b = −0.02, *p* = 0.590) were not associated with worry about getting COVID-19.

### 3.8. Perceived COVID-19 Susceptibility

Many participants (*n* = 378, 67%) indicated that they were “extremely likely” (*n* = 90, 16%), “moderately likely” (*n* = 132, 23%), or “slightly likely” (*n* = 156, 28%) to get infected by COVID-19 in their lifetime. About 8% (*n* = 45) of our participants chose “extremely unlikely”, 13% (*n* = 73) chose “somewhat unlikely”, and 12% (*n* = 67) chose “neither likely nor unlikely” to this question. The mean score of perceived COVID-19 susceptibility was 3.94 (*SD* = 1.49). AI/AN women reported lower perceived COVID-19 susceptibility than men (b = −0.43, *p* = 0.001). Being older (b = 0.05, *p* < 0.001) and having a lower education (b = −0.13, *p* = 0.009) were associated with higher perceived susceptibility.

### 3.9. Perceived COVID-19 Severity

The actual scores for perceived COVID-19 severity ranged from 7 to 20 (M = 13.18, *SD* = 2.20). AI/AN women reported lower perceived severity than men (b = −0.75, *p* < 0.001). Being older was associated with higher perceived severity (b = 0.09, *p* < 0.001). Education was not associated with perceived severity (b = 0.06, *p* = 0.408).

### 3.10. Unadjusted Linear Regression Models

Our sets of bivariate linear regressions indicated that those who had higher NVS health literacy scores (b = 0.10, *p* < 0.001) and higher mental health literacy (b = 0.05, *p* < 0.001) had a higher self-rated physical health status (b = 0.27, *p* < 0.001) and a higher self-rated mental health status (b = 0.28, *p* < 0.001), were more worried about getting COVID-19 (b = 0.36, *p* < 0.001), had higher perceived susceptibility (b = 0.27, *p* < 0.001), and had higher perceived severity (b = 0.20, *p* < 0.001), where they tended to be more willing to receive a COVID-19 vaccine.

AI/AN people’s mental health literacy predicted 30.90% of the variance (R^2^_adjusted_) in the willingness to receive a COVID-19 vaccination. Perceived COVID-19 severity predicted 16.78% of the variance in the willingness to receive a COVID-19 vaccination. Perceived susceptibility predicted 14.48% of the variance in the willingness to receive a COVID-19 vaccination. Being worried about getting COVID-19 predicted 12.60% of the variance in the willingness to receive a COVID-19 vaccination. The self-rated physical health status and mental health status predicted 6.52% and 8.83% of the variance in the willingness to receive a COVID-19 vaccination, respectively. The NVS health literacy scores predicted 4.65% of the variance in the willingness to receive a COVID-19 vaccination.

### 3.11. Adjusted Linear Regression Model

As shown in Table 2, our adjusted linear regression model indicated that after holding the self-rated physical/mental health status, worry about getting COVID-19, perceived susceptibility, perceived severity, NVS health literacy scores, and socio-demographics (i.e., gender, age, and education) constant, mental health literacy was a significant predictor (b = 0.03, *p* < 0.001) of the willingness to receive a COVID-19 vaccination (R^2^_adjusted_ = 40.14%). Those with higher mental health literacy were more willing to receive a COVID-19 vaccination. Those with higher perceived COVID-19 severity and being more worried about themselves getting COVID-19 were also more willing to receive a COVID-19 vaccination (both *p* < 0.001). Interestingly, we found that those with lower NVS health literacy scores were more willing to receive a COVID-19 vaccination (b = −0.04, *p* = 0.035) when holding the other variables constant. Regarding multicollinearity, none of the predictors exhibited a VIF exceeding 2.34, signifying that multicollinearity is not a significant concern in this dataset [44].

## 4. Discussion

We identified mental health literacy as a strong predictor of being willing to receive a COVID-19 vaccination on top of other key factors including socio-demographics, self-rated physical and mental health status, worry about getting COVID-19, perceived COVID-19 susceptibility, and perceived COVID-19 severity among AI/AN persons. AI/AN individuals with higher mental health literacy score were more willing to receive a COVID-19 vaccine. Our findings indicate that implementing educational interventions to improve mental health literacy might be an effective strategy to reduce COVID-19 vaccine hesitancy among AI/AN individuals and communities [45].

During the middle of the COVID-19 pandemic (August 2020–February 2021), the percentage of adults reporting that they needed, but did not receive, mental health counseling or therapy increased significantly in the U.S. (from 9% to 12%) [46]. Thus, there is a critical need for mental health literacy intervention among AI/AN people. Educational interventions increasing knowledge about the warning signs of mental health concerns and mental illness can enhance people’s mental health literacy [47]. Launching community campaigns to raise understanding of mental health challenges, implementing programs in educational settings such as high schools and universities, training members of the public in how to provide mental health first aid support for those who need help, and developing web-based mental health supports and interventions are known to work in improving mental health literacy [48]. It is essential to provide evidence-based culturally informed and culturally relevant interventions to improve mental health literacy among AI/AN people. Educational materials prioritizing Indigenous cultures and values as well as using phrases and words from Indigenous community members enhance the effectiveness of mental health literacy interventions [49].

Another important finding in our study was that higher health literacy was associated with being more willing to get vaccinated against COVID-19, which is consistent with previous studies [16,17]. Interestingly, after holding the self-rated physical/mental health status, worry about getting COVID-19, perceived susceptibility, perceived severity, mental health literacy, and socio-demographics (i.e., gender, age, and education) constant, AI/AN people who reported higher NVS health literacy scores were less likely to be willing to receive a COVID-19 vaccine. This unexpected finding was due to the fact that more than half of our participants (54%) either did not know the answers to the NVS test or simply refused to answer the questions (i.e., skipped the entire NVS measure), which led to getting 0 scores. The NVS mean score of our sample was 1.96 (*SD* = 2.26). The NVS mean score of a U.S. nationally representative adult sample was 4.73 (*SD* = 1.24) for metro residents and 2.99 (*SD* = 2.26) for non-metro residents, respectively [15]. Many participants were unwilling to engage in the required cognitive process to answer NVS questions, which contributed to the low NVS mean score among our sample.

We found that AI/AN women were less likely to be willing to receive a COVID-19 vaccination compared to AI/AN men, which is consistent with prior studies [20,50] because women often exhibit higher levels of hesitancy, driven by concerns regarding the vaccine effectiveness and the fear of injection [51]. We also found that being older was associated with more willingness to receive a COVID-19 vaccine, which is again consistent with previous research [19,52,53]. This might be due to the fact that older people tend to be more worried about getting COVID-19 and hold higher levels of perceived susceptibility and severity of COVID-19 compared to younger people [54], which in turn makes them more willing to receive a COVID-19 vaccination. Also, younger people often exhibit lower overall awareness of vaccinations and diseases, which contributes to their higher level of COVID-19 vaccine hesitancy [20].

### Limitations and Future Research

Our findings do not allow for the establishment of a causal relationship due to the use of a cross-sectional research design. Also, our findings may not generalize well to a larger AI/AN population, especially those who live in rural areas and have low socio-economic status, because of the nature of the study participants recruited to fill out our online survey. Our data collection was also conducted before the COVID-19 vaccine became widely available in the U.S. Recent data indicates that AI/AN communities have high COVID-19 vaccination rates [55]. The success of the COVID-19 vaccination initiative in AI/AN communities can be attributed to its strong foundation, which is deeply rooted in community resilience and a profound respect for tribal sovereignty [56]. Future research should focus on the requirements for vaccination communication and explore the potential contributions of culturally aligned communication campaigns to elevate COVID-19 vaccination rates [57]. Although the NVS health literacy measure elicits reliable data from our AI/AN sample, its validity may be constrained due to the fact that four out of the six questions necessitate numerical calculations directly depending on strong numeracy skills [58], which may not be exhibited in this sample. Health literacy is a multidimensional concept [59,60], with over 50 available measures to assess an individual’s health literacy [61]. Using different health literacy measures could produce different results because these measures may assess varied skill sets. Subsequent studies could assess an individual’s health literacy by using a dedicated COVID-19 vaccine literacy measure [62,63], aiming to enhance our understanding of the connection between health literacy and COVID-19 vaccination behaviors. In addition, our self-reported survey results may exhibit bias when compared to studies employing objective measures. Further qualitative and quantitative research is warranted to understand the reasons why, compared to AI/AN men, AI/AN women reported less willingness to receive a COVID-19 vaccination, lower mental health literacy, lower perceived COVID-19 susceptibility and severity, as well as a poorer self-rated physical and mental health status. Last, future research may examine the influence of perceived barriers and benefits of getting a COVID-19 vaccine on predicting the willingness to receive a COVID-19 vaccine among AI/AN people. We did not measure these constructs in our current study. Previous research reported that perceived barriers and perceived benefits were the most common Health Belief Model constructs associated with vaccine hesitancy [37]. It would also be valuable to compare the findings in this study conducted with the AI/AN people with other depressed populations born in the United States, such as some at-risk African American, Latino American, and White American sectors of the population.

## 5. Conclusions

Our study contributes to the literature by clarifying the effects of mental health literacy and health literacy on COVID-19 vaccine acceptance among AI/AN persons. We conclude that a higher level of mental health literacy strongly predicts the willingness to receive a COVID-19 vaccine within this specific population. Our study raises concerns that AI/AN individuals with limited mental health literacy may be less motivated than other people to receive a COVID-19 vaccine, which increases their vulnerability to infection and the associated physical and mental health challenges during the pandemic. Developing and implementing educational interventions to improve mental health literacy might be an effective strategy to reduce COVID-19 vaccine hesitancy among AI/AN people.

## Figures and Tables

**Table 1 vaccines-11-01793-t001:** Participants’ demographic characteristics (*n* = 563).

Socio-Demographics	N or Mean	% or Range
Age	33.81	20–66
Gender		
Men	347	61.63
Women	215	38.19
Missing	1	0.18
Education		
8 through 11 years	44	7.82
12 years or completed high school	60	10.66
Post-high school training other than college (vocational or technical)	74	13.14
Some college	247	43.87
College graduate	117	20.78
Postgraduate	19	3.37
Missing	2	0.36

**Table 2 vaccines-11-01793-t002:** Predictors of willingness to receive a COVID-19 vaccine among AI/AN population.

Predictors	b	SE	95% CI	*p*
Mental health literacy	0.03	0.004	[0.02, 0.04]	<0.001 **
Perceived COVID-19 severity	0.11	0.02	[0.07, 0.14]	<0.001 **
Perceived COVID-19 susceptibility	0.05	0.03	[−0.02, 0.11]	0.144
Worry about getting COVID-19	0.22	0.04	[0.14, 0.30]	<0.001 **
Self-rated physical health status	−0.06	0.05	[−0.15, 0.03]	0.189
Self-rated mental health status	0.08	0.04	[−0.004, 0.16]	0.062
NVS health literacy score	−0.04	0.02	[−0.08, −0.003]	0.035 *
Age	0.01	0.01	[−0.005, 0.02]	0.191
Sex: women	−0.10	0.08	[−0.26, 0.05]	0.184
Education	0.04	0.03	[−0.02, 0.10]	0.219
Model Adjusted R^2^ = 40.14%				

Note. SE = standard error; CI = confidence interval; * *p* < 0.05, ** *p* < 0.001.

## Data Availability

The data are not publicly available due to privacy or ethical restrictions.

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
