# Peer review of "Identifying Mental Health Literacy as a Key Predictor of COVID-19 Vaccination Acceptance among American Indian/Alaska Native/Native American People"

_vaccines, 2023, doi:10.3390/vaccines11121793_

Round 1

Reviewer 1 Report

Comments and Suggestions for Authors

The idea of the project is interesting. However, I cannot determine why they take two ethnic populations that according to the literature do not have the same opportunities as Native Americans. Within the Native Americans are depressed populations such as some African American, Latino American, and White American sectors.  So making assertions of the type made, it seems to me, is not valid, since they do not compare it to a literature that evaluates other depressed populations born in the United States.

Although all projects do not have to have a control group, the statements in the results could be hypotheses for new work, but they seem a little far from reality.

For example, when they did the experimental design  in the Health literacy: How true is it that other American populations of low, medium or high educational levels are able to interpret mock-up ice-cream nutrition label?

The same goes for the Mental health literacy questions:A person with schizophrenia may see things that are not really there? Do other depressed populations in the U.S. know what schizophrenia is?

And self-rated physical and mental health status by asking, "In general, would you say your physical/mental health is...", Mental health has always been taboo, especially for men. Wouldn't it be that women answered with more and not that they were sicker? Assertions such as: Our findings indicate that the implementation of educational interventions to improve mental health literacy could be an effective strategy to reduce hesitancy about COVID-19 vaccination among AI/AN individuals and communities.......... are statements that cannot be made, the researchers do not know the population, have not asked their needs, have not looked at other populations in the same conditions, and do not have results from other population groups that suffer racism, exclusion, etc., the same as these populations suffer.

In conclusion: I find the conclusions too risky, first because theydo not  compare with other at-risk populations: secpond because the questions they ask do not seem to account for what they want to know, since mental health is a very difficult subject to assess, because of the questions they asked and just because being  "crazy", which is what people think they are asking, is not something that is answered with sincerity.

Reviewer 2 Report

Comments and Suggestions for Authors

 In an easy-to-follow introduction the authors present the background for this interesting study that sought to identify key predictors of COVID-19 vaccination acceptance among American Indian/Alaskan Native, Native American people. The introduction presents a fair cross section of the relevant background literature enabling the reader to appreciate why this study was undertaken. 

The introduction concludes with a three-point summary of the critical research questions that were to be investigated.

Methods: The paper clearly describes the background for the design and use of appropriate questions enabling a reliable estimate of self-rated health literacy, mental health status and literacy. The data collection was carried out employing an accepted valid sampling strategy that sent recruitment flyers to colleagues who conduct research or practice among AI/AN communities. Those professionals then further disseminated the recruitment information to AI/AN people within their social/professional networks. A clear description of the dissemination of the survey instruments is important to enable the reader to gauge the validity of the technique employed once the construction of the survey instruments had been agreed. This appears to have satisfied the requirements of valid surveys.

The number of participants for the study appears to be satisfactory, total participants 563 AI/AN, with a M to F ratio of 62% to 38%.

For each of the predictor variables, adequate care has been taken to determine good internal consistency with the objective test the Newest Vital Sign (NVS) being employed to assess health literacy. The assessment of mental health literacy is also adequately described with examples.

Several covariates were chosen for inclusion in the survey. Some of the chosen covariates had previously been shown to have a relationship with vaccine hesitancy. The physical and mental health status was self-rated using a five-point Likert scale. Concern about contracting COVID-19 was also assessed as was perceived severity of contracting COVID-19.

Overall, the variables selected for inclusion in the survey were justified based on relationship to COVID-19 from previous studies on different ethnic groups.

Data Analysis: This was carried out employing standard statistical analysis methodology that was appropriate for a dataset of this type.

Results: The demographic characteristics of the participants is quite clearly summarised in Table 1.

Under sub-headings 3.2 to 3.8 the authors summarise their data treating each of the covariates to reveal a number of interesting and significant relationships eg AI/AN women were less likely to be willing to receive a COVID-19 vaccination compared to men. AI/AN women also reported poorer self-rated physical health status compared to men and older age was associated with being more worried about getting COVID-19. Many of these observations are interesting and useful.  Some similar observations have been reported in other studies including studies of vaccine hesitancy in residents of Palermo, Sicily versus those living in Bologna, Italy. A recent study investigating vaccine hesitancy in different sectors of a farming community in Guatemala also comes to mind (Vaccines, Wagner et al, 2019).

The adjusted linear regression model summarised in Table 2 was useful and effective in supporting the relationship between higher mental health literacy and willingness to receive a COVID-19 vaccination. This adjusted model also established that those with lower NVS health literacy scored were more likely to be willing to receive a COVID-19 vaccination when other variables are held constant. The benefits of using an adjusted linear regression model are indicated and it appears that these statistical manipulations have been carried out reliably.

Table 2 is very effective in summarising predictors of willingness to receive COVID-19 vaccination among the AI/AN population.

The discussion again emphasises the fact that mental health literacy is a strong predictor of willingness to receive COVID-19 vaccination with other key factors being mentioned. The findings of this study are compared with reports from previous studies but the importance of providing evidence-based culturally informed and culturally relevant interventions to improve mental health literacy among AI/AN people cannot be overemphasised.

The limitations of the study are discussed, and these observations are likely to provide very useful guidance for other studies, particularly those that target communities similar to the AI/AN population.

Suggestions and corrections – very few so well done with attention to detail!

L79 … few studies have investigated…

Reviewer 3 Report

Comments and Suggestions for Authors

In this paper authors examined how health literacy is associated with willingness to receive COVID19 vaccines among AI/AN people. In this study they used Health Belief Model and asked specific questions according to the model to the participants. In the participants' demographic characteristics, they included age, gender, and education as variables. My question is when you study Social Determinants of Health, the two important predictors are education, and income. In this article authors did not include individual income in the demographics. Although they mentioned in the article that majority of the participants are poor. I think, if they include "income" in their socio demographics, it will give a strong predictor of the regression analysis.  

Author Response

We thank this reviewer for the valuable feedback. The first and second authors have actively collaborated on the Native Connections grant with the Iowa Tribe of Oklahoma over the last five years. Through these partnerships, we have gained insights into the sensitivity surrounding personal information among Native individuals, particularly concerning income disclosure. Our extensive research and advocacy efforts within various tribal communities have revealed that specific demographic details, such as gender and income level, are perceived as private and sensitive.

We recognize that some tribal groups regard income as a personal and sensitive aspect. Out of respect to Native American people, we decided not to include the question asking for income level in our survey.

Educational attainment serves as a meaningful indicator of socioeconomic status, offering valuable insights without intruding on the personal boundaries that income disclosure might breach. It is a more acceptable and accessible metric, facilitating a more open and respectful dialogue within the research framework.

Based upon the extensive experience of this research group in studying vulnerable and at-risk populations, we made every possible effort to conduct this research in a culturally sensitive manner. Thank you again for your valuable recommendations.